# OpenReview forum: "When Agents “Misremember” Collectively: Exploring the Mandela Effect in LLM-based Multi-Agent Systems"
_ICLR.cc/2026/Conference — ICLR 2026 Poster_

### Official Review · Reviewer_H3cb · 2025-10-27

**Soundness:** 3
**Presentation:** 3
**Contribution:** 3
**Rating:** 8
**Confidence:** 5

**Summary:**

An interesting paper studies a collective cognitive bias in multi-agent LLM systems akin to the human “Mandela effect,” where group interaction and specious evidence induce shared false memories that can persist over time. The authors also introduce MANBENCH to measure this phenomenon.

**Strengths:**

1. Introduces MANBENCH, the first systematic benchmark to evaluate the Mandela effect in LLM-based multi-agent systems, covering tasks, interaction protocols, and metrics.
2. Quantifies how the effect manifests and operates across model families, group compositions, group sizes, memory timescales, and knowledge domains.
3. Proposes a two-layer defenses—prompt engineering and alignment via SFT—with dataset design and training details, and empirically validates their effectiveness.

**Weaknesses:**

1. While the paper proposes two types of mitigation strategies (prompt-level and model-level defenses), these approaches remain relatively preliminary. For instance, the prompt-level methods rely heavily on predefined rules (e.g., cognitive anchoring and source scrutiny), which may not generalize well to unseen cases or adversarial scenarios.

2. The paper would benefit from adding a discussion section to explore the practical risks of the Mandela Effect in real-world tasks, such as sensitive decision-making scenarios.

**Questions:**

See weakness

---

> ### Author Response · Authors · 2025-11-21
> **Response to Reviewer H3cb (1/1)**
>
> We are encouraged by the reviewer's appreciation of our work and your recognition of our contributions. We appreciate insightful and constructive feedback on improving this paper! Please find below our response to the reviewer’s questions.
>
> >**W1.** While the paper proposes two types of mitigation strategies (prompt-level and model-level defenses), these approaches remain relatively preliminary. For instance, the prompt-level methods rely heavily on predefined rules (e.g., cognitive anchoring and source scrutiny), which may not generalize well to unseen cases or adversarial scenarios.
>
> First, we would like to claim that our primary contribution is to be the first to systematically identify, reproduce, and quantify the Mandela effect in LLM-based multi-agent systems (via MANBENCH). Therefore, the main goal of our proposed defenses (**Section 5**) was to demonstrate that this effect can be mitigated and to provide a baseline for future work.
>
> Second, we would like to clarify that the proposed defenses demonstrate a certain level of domain generalizability. Specifically, the rule-based cognitive anchoring and source scrutiny strategies teach the model a general reasoning process (e.g., checking internal knowledge before accepting external input, identifying the burden of proof, and analyzing narrative structure) rather than domain-specific facts. Furthermore, the model-level defense internalizes this critical thinking capability into the model's parameters, rather than relying solely on in-context prompts. Therefore, the proposed defenses may handle unseen cases themselves to some extent.
>
> Finally, we agree that it is important to develop more generalizable and robust defense mechanisms against the Mandela effect. For instance, introducing "critic" agents for cross-verification and reflection, to ensure alignment with factual ground truth and enhance reasoning robustness, may be a more adaptive defense. We have added the above discussion in the future work part (**Lines 527-529**).
>
> >**W2.** The paper would benefit from adding a discussion section to explore the practical risks of the Mandela Effect in real-world tasks, such as sensitive decision-making scenarios.
>
> Following the reviewer’s suggestion, we have added a detailed discussion about the risks of the Mandela Effect in the medical question-answering domain (**Appendix E**), which is a highly sensitive scenario. Specifically, in systems for diagnostic assistance, the consequences could be severe if a group of agents reaches a consensus based on mutually reinforced misinformation (e.g., a misremembered medical symptom). To validate this, we evaluated on a 1,000-question subset of MedMCQA [1], a challenging medical question-answering dataset. As shown in the following table, under the Role-based Short-term (RS) protocol, nearly half of its correct knowledge is overturned ($\sigma^{RS}=41.97\\%$), showing that medical assistance is also a vulnerability.
>
> |Method|$\sigma^{GS}$ (%) ↓|$\sigma^{GL}$ (%) ↓|$\sigma^{RS}$ (%) ↓|$\sigma^{RL}$ (%) ↓|
> |---|---|---|---|---|
> |MedMCQA|35.08|29.84|41.97|32.13|
>
> We also provide some real cases in Appendix E.2 to further demonstrate the practical risks. For example, in Case 2 (**Appendix E.2.2**), when diagnosing a patient with myositis ossificans progressive, the vanilla model correctly identified "Lung disease" as the usual cause of death. However, under the social influence of a group confidently arguing for "Septicemia" based on plausible but flawed reasoning (e.g., linking immobility to infection risks), the agent abandoned its correct medical knowledge and aligned with the fatal misdiagnosis. This illustrates how the Mandela effect can lead to consensus-based errors in high-stakes decision-making.
>
> [1] MedMCQA: A Large-scale Multi-Subject Multi-Choice Dataset for Medical domain Question Answering, PMLR 2022.
>
> Again, we thank the reviewer for the valuable feedback. Please let us know if there are any other questions or suggestions.

---

### Official Review · Reviewer_LqGM · 2025-10-27

**Soundness:** 3
**Presentation:** 2
**Contribution:** 2
**Rating:** 2
**Confidence:** 1

**Summary:**

This paper investigates the "misremember" effects that could potentially happen among multi-agent systems. It involves repurposing an existing QA benchmark and investigating several settings where agents could be wrong.
Evaluation with several current SOTA LLMs shows that all suffer from such effects.
Furthermore, this work investigates prompting and sft methods to defend models against such effects and shows improvement.

**Strengths:**

I agree that multi-agent systems are increasingly involved in complex workflows, and investigating the failure case of a multi-agent system is an important issue. This paper focuses on a specific phenomenon, proposes an evaluation, and shows improvement methods. The findings align with expectations, and the ablations show more insight into the failure of mult-agent systems in LLM era.

**Weaknesses:**

Overall, I’ve seen many recent works draw on psychological concepts from human society and apply them to multi-LLM agent systems. However, we should keep in mind that LLMs differ fundamentally from humans; they have near-perfect memory, remain largely homogeneous, and are trained to play sycophantic roles as “user” and “assistant,” making them naturally inclined to agree with users. Specifically, I have several concerns about the work.

First, the idea of a specific malicious agent that may render the whole multi-agent system down is not new, for example, https://arxiv.org/abs/2408.00989 (btw, this work is not cited in the paper). Although the specific topics might slightly differ from each other, I find it hard to tell how the evaluation framework is fundamentally different from previous work.

Second, the contribution of the dataset seems to be simply repurposing an existing dataset, and I would argue that such context might not be the most practical scenario where the ground truth is aware to each party. That being said, the proposed benchmark is probably not a challenging benchmark, as each model would actually have the correct answer, and the most straightforward approach is to simply prompt every agent not to listen to each other. This hypothesis seems to be verified by the authors' later effort in climbing on the benchmark. The real challenge, however, is for scenarios where agents have to collaborate with each other to get information or knowledge. Can authors think of more scenarios in this direction?

Third, the experiments are only done with a single LLM for each simulation. Do authors have plans to run multiple LLMs in one scenario? Especially, this setting would be more aligned with the human society Mandela effect, where each party has different knowledge.

**Questions:**

see Weakness.

---

> ### Author Response · Authors · 2025-11-21
> **Response to Reviewer LqGM (1/3)**
>
> We thank the reviewer for the insightful and constructive feedback on improving this paper! Please find below our response to the reviewer’s questions.
>
> > **W1.** Overall, I’ve seen many recent works draw on psychological concepts from human society and apply them to multi-LLM agent systems. However, we should keep in mind that LLMs differ fundamentally from humans; they have near-perfect memory, remain largely homogeneous, and are trained to play sycophantic roles as “user” and “assistant,” making them naturally inclined to agree with users.
>
> We agree that LLMs differ fundamentally from humans, but psychology-inspired research has been crucial in enhancing LLM multi-agent systems. For instance, **hallucination** [1] uses human perceptual distortion to fix factual errors, and **sycophancy** [2] leverages human social compliance theories to reveal alignment issues. Inspired by the Mandela effect, our work explores how social dynamics create collective false memories.
>
> Crucially, the specific LLM traits you mentioned (near-perfect memory, homogeneity, sycophancy) do not protect against the Mandela effect; our data shows they actually facilitate it:
>
> - LLM memory is both **factually imperfect** and highly **susceptible to social distortion**. Our Baseline Reality Protocol shows that even SOTA models are not perfect, with GPT-5 exhibiting a 17.63% error rate. Furthermore, even when memory is initially correct, it is fragile and susceptible to social influence. For instance, GPT-5's error rate more than doubles from 17.63% to 41.59% under the Role-based Short-term (RS) protocol.
>
> - Homogeneity does **not** prevent the Mandela effect. We observe that agents with the same knowledge still form false memories. As shown in the reality shift rate of **13 representative LLMs** from Table 3 (**Lines 275-288**), agents with identical training data abandon their initial knowledge to align with false consensus.
>
> - Trained sycophancy precisely causes the Mandela effect. As the reviewer observes, trained agreeableness makes agents inherently vulnerable to the false consensus. Our work quantifies this risk.
>
> [1] Survey of Hallucination in Natural Language Generation, ACM Computing Surveys 2023.
>
> [2] Towards Understanding Sycophancy in Language Models, ICLR 2024.
>
>
> > **W2.** First, the idea of a specific malicious agent that may render the whole multi-agent system down is not new, for example, https://arxiv.org/abs/2408.00989 (btw, this work is not cited in the paper). Although the specific topics might slightly differ from each other, I find it hard to tell how the evaluation framework is fundamentally different from previous work.
>
> We thank the reviewer for highlighting this insightful work [1], which provides a valuable analysis of multi-agent robustness against faulty agents. We have now cited it in our revised Related Work (Lines 122-125).  While both [1] and our work involve misinformation in multi-agent systems, we respectfully clarify that they differ fundamentally across three dimensions: **research objectives**, **implementation methods**, and **evaluation findings**.
>
> - **Research objectives:** We focus on cognitive vulnerability rather than fault tolerance.
>
>   - [1] adopts a system engineering perspective to prevent faulty agents from breaking the system's functionality.
>
>   - We explore how social dynamics induce agents to share and internalize a false memory. We aim to reveal a specific **cognitive bias** (the Mandela Effect) where agents actively collaborate to construct a false reality, rather than a random functional breakdown.
>
> - **Implementation methods:** We focus on complex social engineering rather than a single faulty agent injection.
>
>   - [1] employs a malicious agent that directly introduces faults.
>
>   - Our Role-based Protocols (RS/RL) simulate social engineering by groups of agents with social roles (e.g., Authority Endorser, Questioning Compromiser), collectively constructing a plausible but false narrative.
>
> - **Evaluation findings:** We focus on memory internalization more than just in-context influence.
>
>   - [1] evaluates immediate, in-context impacts to check if the system answers incorrectly during the interaction.
>
>   - We measure both in-context influence through our Short-term Protocols (GS/RS) and memory internalization through our Long-term Protocols (GL/RL). We find that agents are affected not only by short-term sycophancy (agreeing in context) but also by long-term false memory (retaining the false belief in isolation).
>
> [1] On the Resilience of LLM-Based Multi-Agent Collaboration with Faulty Agents, ICML 2025.

---

> ### Author Response · Authors · 2025-11-21
> **Response to Reviewer LqGM (2/3)**
>
> > **W3.** Second, the contribution of the dataset seems to be simply repurposing an existing dataset, and I would argue that such context might not be the most practical scenario where the ground truth is aware to each party. That being said, the proposed benchmark is probably not a challenging benchmark, as each model would actually have the correct answer, and the most straightforward approach is to simply prompt every agent not to listen to each other. This hypothesis seems to be verified by the authors' later effort in climbing on the benchmark. The real challenge, however, is for scenarios where agents have to collaborate with each other to get information or knowledge. Can authors think of more scenarios in this direction?
>
> We thank the reviewer for this critical observation. We strongly agree that the ultimate goal of multi-agent systems is collaborative knowledge acquisition. However, we respectfully posit that the "ignore others" strategy is detrimental to multi-agent systems' utility, and that our benchmark tests a necessary prerequisite for successful collaboration.
>
> -  **Simply ignoring others undermines agents' collaboration.** The reviewer suggests "simply prompt every agent not to listen to each other". While this prevents the Mandela effect, it negates the purpose of a multi-agent system, which is to leverage collective intelligence.
>
>    - **Collaboration is necessary in multi-agent systems.** In multi-agent systems, agents are often designed to achieve collective goals through group discussion, reasoning, and argumentation. However, if agents ignore each other's opinions, they cannot truly collaborate. This would reduce them to isolated single-agent models. For example, in **scientific research** or **medical diagnosis**, synthesizing multiple expert opinions is essential rather than outright dismissing them.
>
>    - **Simple isolating agents hurts overall utility.**  Fine-tuning LLMs to **simply "not listen"** (using only the resilience set) causes them to **exclude other agents' answers unconditionally, whether right or wrong**. For instance, the reality shift rate on even helpful inputs spiked to 38.5% (under the Correct Guidance Protocol) for Llama3.1-8B (**Figure 6**, **Line 483**). Therefore, in agent collaboration, deciding which content from other sources to incorporate and which to ignore for each agent is complex. A robust agent must possess discernment to accept truth and reject falsehood.
>
> - The challenge is **not** retrieving the fact, but **maintaining that fact against social influence**. The reviewer notes that models "actually have the correct answer." We argue that this is precisely what makes the benchmark revealing. The definition of the Mandela effect is not "not knowing," but "misremembering" known facts under social influence. As shown in Table 3 (**Line 275**), the fact that agents abandon their correct internal knowledge (e.g., GPT-5's error rate doubles from 17.63% to 41.59% under influence) proves that possessing the correct answer is insufficient protection against social manipulation.
>
> - MANBENCH is **not** simply the repurposing of existing QA benchmarks, but creating a **simulation framework**, which includes (i) novel interaction protocols that generate social dialogue turns rather than static QA; (ii) new evaluation metrics (e.g., reality shift rate) to quantify memory persistence; and (iii) diverse tasks susceptible to the Mandela effect.
>
> - **Future work.** We agree with the reviewer that a more challenging scenario is agents "collaborate to get information". However, before we trust agents to collaborate to discover unknown knowledge, we must first ensure **they possess cognitive resilience to avoid distorting the known facts** they already possess. MANBENCH provides the necessary testing ground for this foundational capability. Our future work will focus on (i) expanding MANBENCH to include more challenging benchmark tasks that require cooperation from each agent to complete, and (ii) exploring more interaction protocols that better mimic real-world collaborative environments (e.g., long-form debate, group discussion, and collaborative planning).  We have added the relevant limitation (**Lines 513-521**) and are committed to addressing this in future work (**Lines 522-529**).

---

> ### Author Response · Authors · 2025-11-21
> **Response to Reviewer LqGM (3/3)**
>
> > **W4.** Third, the experiments are only done with a single LLM for each simulation. Do authors have plans to run multiple LLMs in one scenario? Especially, this setting would be more aligned with the human society Mandela effect, where each party has different knowledge.
>
> We thank the reviewer for this insightful suggestion. We agree that real-world social groups (and the human Mandela effect) involve diverse participants with varying knowledge bases. To address this, we have conducted **new experiments with heterogeneous groups** and added the results to the Discussion section (**Lines 498-505**) and **Appendix B**.
>
> - Rationale for Homogeneous Baseline
>
>   In our main experiments, we used homogeneous agents (identical underlying LLMs) for two key reasons:
>
>   - **Controlled Variable:** It measures the impact of the social dynamic without variables from models' knowledge.
>
>   - **Industrial Relevance:** Most current multi-agent systems (e.g., CAMEL [1], AgentVerse [2], and MetaGPT [3]) rely on a single LLM backbone to drive all agents. Thus, homogeneity represents a critical "threat model" for current deployments.
>
> - New Experiments: Heterogeneity Amplifies the Effect
>
>   - Following the reviewer's suggestion, we further construct a heterogeneous scenario to evaluate interactions between LLMs of different types. Recall that in the homogeneous setting, all agents are based on the subject agent. Comparatively, in the heterogeneous setting, a single subject agent (the one providing the final answer) is influenced by a group of five other distinct models. The heterogeneous group comprises Qwen3-235B-A22B, GPT-5, Claude-Sonnet-4, Gemini-2.5-Pro, Llama-3.3-70B, and Deepseek-v3.1.
>
>   - **Key finding.** We observe that heterogeneity **amplifies** the Mandela effect due to perceived capability differences and more diversity narratives (as supported by higher reality shift rates $\sigma$).
>
> |Subject Agent|Experiment Setting|$\sigma^{GS}$ (%) ↓|$\sigma^{GL}$ (%) ↓|$\sigma^{RS}$ (%) ↓|$\sigma^{RL}$ (%) ↓|
> |---|---|---|---|---|---|
> |Qwen3-235B-A22B|Homogeneous|66.98|47.65|68.69|56.85|
> |Qwen3-235B-A22B|**Heterogeneous**|69.75|58.20|78.67|65.43|
> |GPT-5|Homogeneous|27.42|2.96|31.03|1.67|
> |GPT-5|**Heterogeneous**|28.54|3.41|35.82|2.24|
>
> Again, we thank the reviewer for the valuable feedback. Please let us know if there are any other questions or suggestions.
>
> [1] CAMEL: Communicative Agents for "Mind" Exploration of Large Language Model Society, NeurIPS 2023.
>
> [2] AgentVerse: Facilitating Multi-Agent Collaboration and Exploring Emergent Behaviors, ICLR 2024.
>
> [3] MetaGPT: Meta Programming for A Multi-Agent Collaborative Framework, ICLR 2024.

---

### Official Review · Reviewer_ABwn · 2025-10-28

**Soundness:** 3
**Presentation:** 3
**Contribution:** 3
**Rating:** 6
**Confidence:** 4

**Summary:**

This paper investigates the Mandela effect (collective false memory) in LLM-based multi-agent systems, a critical yet underexplored issue in collaborative AI. The authors introduce MANBENCH, a novel benchmark with 4,838 questions across 4 task domains (e.g., History, Domain-Specific Knowledge) and 5 interaction protocols (e.g., Role-based Short-term/Long-term) to evaluate the phenomenon. They test 13 LLMs (7 commercial, 6 open-source), confirming all models are susceptible to the Mandela effect—for instance, Qwen3-235B’s error rate rises from 25.48% (baseline) to 74.75% under the Role-based Short-term Protocol.

**Strengths:**

+ The idea proposed in this paper is interesting and insightful.
+ This study is the first to systematically explore collective false memory (Mandela effect) in multi-agent systems, addressing a critical gap between individual LLM hallucination and group-level cognitive biases.
+ The paper provides comprehensive experiments and analysis, including diverse interaction protocols (simulating short/long-term memory and generic/role-based groups), and tailored metrics (reality shift rate, σ_max) etc.

**Weaknesses:**

- Although MANBENCH tasks are adapted from BIG-Bench Hard, they may not fully reflect the complexity of real-world multi-agent interactions (e.g., dynamic role changes, unstructured dialogue), potentially limiting the ecological validity of results.

- While defenses reduce the Mandela effect on MANBENCH, the paper provides little evidence of their performance across unseen tasks or domains (e.g., highly specialized fields like healthcare), leaving uncertainty about their broader applicability.

-  The interaction protocols predefine agent roles (e.g., Error Conclusion Initiator) and consensus direction, but real multi-agent systems often involve uncoordinated, conflicting inputs—this simplification may underestimate or distort how the Mandela effect emerges naturally.

**Questions:**

1. For the model-level defense, how does the balance between the resilience set and cooperative set (e.g., ratio adjustments) impact performance? Could a dynamic ratio (tailored to task domains) further improve both error resistance and knowledge absorption?
2. The paper notes that some models (e.g., GPT-5) self-correct false memories long-term, while others (e.g., Claude 3.5 Haiku) do not. What underlying LLM characteristics (e.g., context window size, training data) drive this difference in memory integrity?
3. MANBENCH focuses on verifiable factual tasks—would the Mandela effect manifest differently in subjective or creative tasks (e.g., collaborative content generation), and how might the proposed defenses adapt to such scenarios?
4. For role-based groups, the "suspicion-induced vigilance" effect reduces the Mandela effect when group size exceeds 9. Does this threshold vary across LLM types (e.g., open-source vs. commercial) or knowledge domains, and can this effect be proactively leveraged in defense design?

---

> ### Author Response · Authors · 2025-11-21
> **Response to Reviewer ABwn (1/3)**
>
> We thank the reviewer for the positive assessment. We appreciate insightful and constructive feedback on improving this paper! Please find below our response to the reviewer’s questions.
>
> ### Response to Weakness:
>
> >**W1.** Although MANBENCH tasks are adapted from BIG-Bench Hard, they may not fully reflect the complexity of real-world multi-agent interactions (e.g., dynamic role changes, unstructured dialogue), potentially limiting the ecological validity of results.
>
> We thank the reviewer for this point. Our primary contribution is to fill a critical research gap by investigating the specific cognitive bias of the Mandela effect (collective false memory) that emerges in LLM-based multi-agent systems. We agree that using multiple-choice questions simplifies the complexity of real-world multi-agent interactions. We selected a multiple-choice question format, which is standard in factuality benchmarks (e.g., BBH [1], MMLU [2] datasets) to ensure **valid and reliable measurement**. In a controlled experimental setting (e.g., interaction type and group composition), this format is crucial for MANBENCH because it allows us to objectively **quantify a reality shift** with a binary change from a correct to an incorrect answer with **verifiable ground truth**.
>
> We agree that real-world interactions are far more complex, involving dynamic roles and unstructured dialogue, which present greater diversity and are more difficult to control. However, we believe our interaction protocols (e.g., Role-based Group) are foundational and can be adapted by future work to study these complex scenarios. Our future work will focus on (i) expanding MANBENCH to include more challenging benchmark tasks that require cooperation from each agent to complete, and (ii) exploring more interaction protocols that better mimic real-world collaborative environments (e.g., long-form debate, group discussion, and collaborative planning).  We have added the relevant limitation (**Lines 513-521**) and are committed to addressing this in future work (**Lines 522-529**).
>
> [1] Beyond the Imitation Game: Quantifying and extrapolating the capabilities of language models, TMLR 2023.
>
> [2] Measuring Massive Multitask Language Understanding, ICLR 2021.
>
> >**W2.** While defenses reduce the Mandela effect on MANBENCH, the paper provides little evidence of their performance across unseen tasks or domains (e.g., highly specialized fields like healthcare), leaving uncertainty about their broader applicability.
>
> We thank the reviewer for highlighting the importance of domain generalization. Our defenses teach the model to develop a **reasoning strategy** (e.g., checking internal knowledge before accepting external input, and analyzing narrative structure) to mitigate the Mandela effect without relying on domain-specific knowledge.
>
> To validate this, we evaluated our defenses on a 1,000-question subset of MedMCQA [1], a challenging medical question-answering dataset. We applied prompt-level defense to the closed-source model and model-level defense to the open-source model and calculated the reality shift rate ($\sigma$).
>
> - Prompt-level defense on GPT-4o-mini. Under the Role-based Short-term (RS) protocol, nearly half of its correct knowledge is overturned ($\sigma^{RS}=41.97\\%$). Applying cognitive anchoring and source scrutiny reduces this shift to 15.41% and 19.34% respectively.
>
> |Model Setting|$\sigma^{GS}$ (%) ↓|$\sigma^{GL}$ (%) ↓|$\sigma^{RS}$ (%) ↓|$\sigma^{RL}$ (%) ↓|
> |---|---|---|---|---|
> |Base (No Defense)|35.08|29.84|41.97|32.13|
> |w/ Cognitive Anchoring|16.39|14.75|15.41|14.42|
> |w/ Source Scrutiny|20.66|15.41|19.34|15.08|
>
> - Model-level defense on Llama 3.1-8B. Under the RS protocol, the reality shift rate is 43.65%. After SFT with our balanced dataset, the model shows increased resilience, stabilizing the reality shift rate at ~23%.
>
> |Model Setting|$\sigma^{GS}$ (%) ↓|$\sigma^{GL}$ (%) ↓|$\sigma^{RS}$ (%) ↓|$\sigma^{RL}$ (%) ↓|
> |---|---|---|---|---|
> |Base (No Defense)|39.15|34.92|43.65|37.30|
> |SFT (Resilience + Coop)|20.74|16.56|23.15|15.43|
>
> We have included these results to show that the defense generalizes to unseen domains (**Appendix E**).
>
> [1] MedMCQA: A Large-scale Multi-Subject Multi-Choice Dataset for Medical domain Question Answering, PMLR 2022.

---

> ### Author Response · Authors · 2025-11-21
> **Response to Reviewer ABwn (2/3)**
>
> >**W3.** The interaction protocols predefine agent roles (e.g., Error Conclusion Initiator) and consensus direction, but real multi-agent systems often involve uncoordinated, conflicting inputs—this simplification may underestimate or distort how the Mandela effect emerges naturally.
>
> We clarify our design choices based on the specific nature of the Mandela effect and red teaming principles.
>
> - On predefined roles. We respectfully highlight that our Generic Group Protocols (GS, GL) address the lack of strategic coordination. In this setting, agents have no predefined roles, simulating a simple social contagion. Our results show that the Mandela effect emerges strongly even without strategic roles, proving that role-play is an amplifier, not a root cause.
>
> - On predefined consensus direction. We follow the real-world Mandela effect where all agents share the same false memories. Predefining the consensus direction is necessary for **causal attribution**. This design ensures we measure the model's resilience against **socially induced errors** (agents were initially correct but became wrong) rather than **coincidental errors** (the agent would have been wrong anyway). We agree that studying uncoordinated inputs is valuable, such as observing agents with conflicting false beliefs to see what collective errors emerge and track how group beliefs shift over interactions.
>
> ### Response to Questions:
>
> > **Q1.** For the model-level defense, how does the balance between the resilience set and cooperative set (e.g., ratio adjustments) impact performance? Could a dynamic ratio (tailored to task domains) further improve both error resistance and knowledge absorption?
>
> Thanks for your suggestion. In our main experiments (**Section 5.2**), we chose a 1:1 ratio of resilience set (to resist false narratives for skepticism) and cooperative set (to accept valid truths for collaboration). We have extended our analysis to different ratios between the resilience set and the cooperative set. The following table shows the reality shift rates for models fine-tuned with different ratios of resilience and cooperative sets, compared with the vanilla model without SFT.
>
> |Model|Dataset Ratio (Res% : Coop%)|$\sigma^{GS}$ (%) ↓|$\sigma^{GL}$ (%) ↓|$\sigma^{RS}$ (%) ↓|$\sigma^{RL}$ (%) ↓|$\sigma^{C}$ (%) ↓|
> |---|---|---|---|---|---|---|
> |Llama 3.1-8B|vanilla model (w/o SFT)|61.7|85.1|99.5|32.1|1.5|
> |Llama 3.1-8B|100% : 0% (Only resilience set)|15.5|12.1|18.2|9.9|38.5|
> |Llama 3.1-8B|75% : 25%|17.9|14.2|19.0|10.8|8.9|
> |Llama 3.1-8B|50% : 50%|18.7|15.3|21.5|11.8|1.1|
> |Llama 3.1-8B|25% : 75%|38.1|31.9|55.2|28.7|0.9|
> |Llama 3.1-8B|0% : 100% (Only cooperative set)|65.3|70.1|99.5|40.1|0.8|
> |Qwen3-8B|vanilla model (w/o SFT)|67.9|50.4|66.8|55.8|1.2|
> |Qwen3-8B|100% : 0% (Only resilience set)|12.3|10.2|14.8|11.5|35.2|
> |Qwen3-8B|75% : 25%|13.8|11.0|15.5|13.2|7.8|
> |Qwen3-8B|50% : 50%|15.1|12.4|17.2|14.3|0.9|
> |Qwen3-8B|25% : 75%|31.5|23.4|28.4|26.0|0.9|
> |Qwen3-8B|0% : 100% (Only cooperative set)|68.5|55.2|67.3|58.1|0.6|
>
> Our results reveal a trade-off between resilience and cooperativeness. A high-resilience ratio (75:25) improves defense performance (\textit{e.g.}, Llama's $\sigma^{RS}$ drops from 21.5\% to 19.0\%) but causes the model to incorrectly reject correct guidance from other agents (\textit{e.g.}, $\sigma^{C}$ increases from 1.1\% to 8.9\%). Conversely, a low-resilience ratio (25:75) maintains the model's ability to accept correct guidance but significantly degrades its defense against misleading information (Llama's $\sigma^{RS}$ rises to 55.2\%). The 50:50 ratio provides an optimal resilience-cooperativeness trade-off, achieving both effective resistance to false memories (low $\sigma^{RS}$) and appropriate acceptance of correct guidance (low $\sigma^{C}$). This demonstrates that the model learns to discriminate between misleading and correct guidance from other agents.
>
> We agree with the reviewer that a dynamic ratio tailored to task domains would further improve performance. For high-ambiguity domains (e.g., History, Medical), the SFT dataset can shift towards a 75%:25% ratio to prevent misinformation. For factual and collaborative domains (e.g., General Knowledge), the SFT dataset can shift towards 25%:75% to promote collaboration. We have added these results to **Appendix C**.

---

> ### Author Response · Authors · 2025-11-21
> **Response to Reviewer ABwn (3/3)**
>
> > **Q2.** The paper notes that some models (e.g., GPT-5) self-correct false memories long-term, while others (e.g., Claude 3.5 Haiku) do not. What underlying LLM characteristics (e.g., context window size, training data) drive this difference in memory integrity?
>
> We thank the reviewer for this insightful question regarding the divergence in long-term memory integrity.
>
> While model implementation details are undisclosed, we hypothesize that this divergence in long-term memory integrity is driven by a fundamental trade-off between **epistemic robustness** and **contextual sensitivity**, rather than simple constraints such as context window size. Models like GPT-5 show high epistemic robustness with strong internal factual checks that override false memories, enabling self-correction. In contrast, models like Claude 3.5 Haiku appear to prioritize contextual sensitivity, allowing false memories to overwrite their original knowledge. This likely results from different alignment strategies: GPT-5 seems to be aligned with **factual truthfulness**, discarding context that conflicts with reality, whereas other models may be aligned with **context adherence** (a desirable trait for RAG or summarization), treating false memories as ground truth.
>
> > **Q3.** MANBENCH focuses on verifiable factual tasks—would the Mandela effect manifest differently in subjective or creative tasks (e.g., collaborative content generation), and how might the proposed defenses adapt to such scenarios?
>
> We thank the reviewer for this forward-looking question. The classic Mandela effect requires a **verifiable ground truth** to define false memory. In subjective tasks (e.g., creative writing), the phenomenon shifts from **false memory** to **homogenization**. In creative tasks, we would expect agents to abandon their unique, long-tail perspectives to converge on **generic or clichéd outputs** rather than diverse, novel content.
>
> Our proposed defenses are structurally adaptable to this context because they target the mechanism of social influence rather than the content itself. Cognitive anchoring (checking facts) would be less effective in subjective domains. However, source scrutiny (analyzing why the group is pushing a narrative) would remain highly relevant. Our cognitive anchoring defense could be adapted by shifting the primacy of the internal knowledge principle to that of the unique perspective, encouraging agents to value diversity over consensus.
>
> > **Q4.** For role-based groups, the "suspicion-induced vigilance" effect reduces the Mandela effect when group size exceeds 9. Does this threshold vary across LLM types (e.g., open-source vs. commercial) or knowledge domains, and can this effect be proactively leveraged in defense design?
>
> Thank you for this insightful observation! We observe that the threshold for the "suspicion-induced vigilance" effect (the group size $N$ where the error rate peaks and begins to drop) varies depending on model types and knowledge domains.
>
> - On model types. Closed-source models tend to exhibit lower vigilance thresholds than open-source models. Specifically, closed-source models like Claude 3.5 Haiku and GPT-4o-mini exhibit a moderate threshold, peaking at $N=6$, suggesting they require a sufficient level of consensus before triggering suspicion. Open-source models appear less sensitive to other agents' opinions. For example, Qwen3-8B triggers vigilance at $N=8$ and Qwen3-32B triggers vigilance at $N=7$.
>
> - On knowledge domains. The threshold varies with the epistemic uncertainty and authority dependence of the knowledge domain. For Claude 3.5 Haiku, in the "General Knowledge" domain, agents have strong internal anchors, so even a small group ($N=4$) of error agents causes suspicion. The "History, Time, & Events" and "Misconceptions & Social Cognition" domains have moderate thresholds ($N=5$ and $N=6$), while "Domain-Specific Knowledge" exhibits the highest threshold ($N=7$). This aligns with our results in **Section 4.3.4** that "Domain-Specific Knowledge" is the most vulnerable to the Mandela effect.
>
> This effect provides a theoretical foundation for our defense design. As discussed in **Section 4.3.3**, the "suspicion-induced vigilance" effect suggests that agents possess a latent capability to detect inauthentic social dynamics. Our source scrutiny defense (**Section 5.1**) proactively uses this mechanism. By explicitly prompting the model to analyze narratives and evaluate credibility, this defense activates the vigilance mechanism without requiring a large group size, enabling the model to identify and reject false memories effectively.
>
> Again, we thank the reviewer for the valuable feedback. Please let us know if there are any other questions or suggestions.

---

### Official Review · Reviewer_BRew · 2025-11-01

**Soundness:** 2
**Presentation:** 3
**Contribution:** 3
**Rating:** 6
**Confidence:** 3

**Summary:**

This paper introduces MANBENCH, a novel benchmark to measure and diagnose the Mandela effect—the formation of shared false memories—in systems of collaborating LLM agents. MANBENCH comprises 4 838 multiple-choice questions drawn from BIG-Bench Hard, organized into four knowledge domains, and five interaction protocols varying in group composition (Generic vs. Role-based) and memory timescale (Short-term vs. Long-term). The authors evaluate 13 state-of-the-art LLMs, quantify a large reality-shift effect across all models, analyze key drivers (group size, domain, model scale, etc.), and propose both prompt-level (cognitive anchoring, source scrutiny) and model-level (supervised fine-tuning with resilience/cooperative data) defenses that reduce the effect by up to 74.4%.

**Strengths:**

1. The problem framing is novel. This paper is the first systematic study of collective false memories in LLM-based multi-agent systems, extending beyond individual hallucinations to social contagion effects.

2. This paper conducts a comprehensive evaluation across 13 models (commercial + open-source) and five protocols, with well-defined metrics (error rate, reality shift rate, maximal shift rate).

3. This paper provides not only an evaluation benchmark but also mitigation methods: two prompt-based strategies and an SFT-based model intervention.

**Weaknesses:**

1. The “specious evidence” dialogues are synthetic. It remains unclear how these engineered narratives map onto real-world multi-agent deployments or user-driven misinformation.

2. The benchmark is limited to multiple-choice questions. The transfer of findings to free-form, open-ended tasks (e.g., long-form debate or planning) is not evaluated.

**Questions:**

1. It is better to include significance tests (e.g., paired bootstrap) to support claims of “significant reduction.”

2. It will be interesting to involve humans in the loop in multi-agent collaboration and to see if the findings still hold. For example, what is the human performance on MANBENCH.

3. Is there any cross-task contamination or transfer of false memories when agents move between distinct domains within the same session?

---

> ### Author Response · Authors · 2025-11-21
> **Response to Reviewer BRew (1/2)**
>
> We thank the reviewer for the positive assessment. We appreciate insightful and constructive feedback on improving this paper! Please find below our response to the reviewer’s questions.
>
> ### Response to Weakness:
>
> > **W1.** The “specious evidence” dialogues are synthetic. It remains unclear how these engineered narratives map onto real-world multi-agent deployments or user-driven misinformation.
>
> We appreciate this important methodological concern and would like to clarify both the rationale for synthetic construction and its connection to real-world scenarios.
>
> - Psychological Grounding: Our interaction protocols are based on psychological findings of the Mandela effect, which demonstrate how collective false memories form among groups [1]. The spread of such memories is linked to social influence processes such as social reinforcement [2]. When exposed to misinformation, individuals' memories can be distorted. Our design mimics these social influence dynamics, with roles like Authority Endorser and Group Consensus Reinforcer aligned with persuasion principles such as authority and social proof [3] used in real-world scenarios [4].
>
> - Controlled Evaluation: Using synthetic, template-controlled narratives allows us to isolate specific variables (e.g., specific role types) to measure their marginal impact on the Mandela effect. With unstructured "in-the-wild" data without synthetic control, we cannot distinguish between **coincidental errors** (the agent would have been wrong anyway) and **socially induced errors** (agents were initially correct but became wrong).
>
> This is analogous to controlled experiments in psychology. While it may differ from real life, it enables rigorous hypothesis testing.
>
> [1] The Visual Mandela Effect as evidence for shared and specific false memories across people, Psychological Science, 2022.
>
> [2] Planting misinformation in the human mind: A 30-year investigation of the malleability of memory, Learning & memory, 2005.
>
> [3] Influence: Science and practice, Boston: Pearson education, 2009.
>
> [4] Network propaganda: Manipulation, disinformation, and radicalization in American politics, Oxford University Press, 2018.
>
> > **W2.** The benchmark is limited to multiple-choice questions. The transfer of findings to free-form, open-ended tasks (e.g., long-form debate or planning) is not evaluated.
>
> First, we would like to explain that the reason we chose the multiple-choice question format, which is standard and most commonly used in factuality benchmarks (e.g., BBH [1], MMLU [2] datasets), is to ensure **valid and reliable measurement**. Specifically, this format allows us to objectively **quantify a reality shift** with a binary change from a correct to an incorrect answer.
>
> Second, we agree that real-world scenarios often involve open-ended tasks where knowledge is uncontrollable, and including such tasks is an important future work. We believe our interaction protocols (e.g., Role-based Group) are foundational and can be adapted to complex, open-ended scenarios. For instance, we plan to adapt our metrics to open-ended debates by measuring shifts in argumentative stance rather than binary answer flips. Similarly, our protocols can be extended to collaborative planning by tracking whether an agent abandons a valid optimal plan for a suboptimal one under peer pressure.  We have added the relevant limitation (**Lines 513-521**) and are committed to addressing this in future work (**Lines 522-529**).
>
> [1] Beyond the Imitation Game: Quantifying and extrapolating the capabilities of language models, TMLR 2023.
>
> [2] Measuring Massive Multitask Language Understanding, ICLR 2021.

---

> ### Author Response · Authors · 2025-11-21
> **Response to Reviewer BRew (2/2)**
>
> ### Response to Questions:
>
> > **Q1.** It is better to include significance tests (e.g., paired bootstrap) to support claims of “significant reduction.”
>
> We appreciate your suggestion to improve our analysis. Following your advice, we performed paired bootstrap tests with 100,000 resamples on the full MANBENCH dataset to validate our mitigation strategies. The error outcomes of the undefended agent served as the baseline, while those of the defended agent (e.g., using Cognitive Anchoring or SFT) were compared. The null hypothesis stated that the defense had no effect or increased errors. Results show a consistent, significant reduction in the Mandela effect across all 13 models. Specifically, GPT-4o and Llama-3.1-8B show p-values less than $1.0×10^{-5}$, allowing us to confidently reject the null hypothesis. The reductions in the reality shift rate ($\sigma^{RS}$) for GPT-4o (from 56.0% to 17.0%) and Llama-3.1-8B (from 99.5% to 18.2%) are both statistically significant.
>
> > **Q2.** It will be interesting to involve humans in the loop in multi-agent collaboration and to see if the findings still hold. For example, what is the human performance on MANBENCH.
>
> We agree that exploring human-in-the-loop dynamics is a vital and fascinating extension of this work. We are actively planning a user study. Due to time constraints of the rebuttal period, we have not yet begun this human-subject research. We are currently preparing our user study protocol for submission to the IRB (Institutional Review Board). This planned study will involve two groups of human participants: one group will answer the MANBENCH questions in isolation to establish a baseline, while the second group will answer the same questions after being exposed to the misleading, specious responses generated by the agent groups. This design will allow us to directly measure the extent to which these agent-generated narratives can influence human judgment and thereby quantify the Mandela effect in human participants.
>
> > **Q3.** Is there any cross-task contamination or transfer of false memories when agents move between distinct domains within the same session?
>
> Thanks for your suggestion. We have implemented a cross-task contamination test procedure using the four knowledge domains identified in our paper. For each of the 12 possible distinct domain pairings (Source $\rightarrow$ Target), we first exposed an agent (GPT-4o-mini) to a Source Domain task and narrative opinions using the Role-based Short-term (RS) protocol. Afterwards, we presented it with a Target Domain task, also under the RS protocol.
>
> We defined a new metric, the **transfer reality shift rate ($\sigma^T$)**, which measures the proportion of questions in the Target Domain ($D _{tgt}$) that were correct at the baseline reality protocol ($B$) but answered incorrectly after being exposed to the questions and narrative opinions from Source Domain ($D _{src}$) in a task transfer protocol ($T _{src \rightarrow tgt}$). This is calculated as $\sigma^T = |\mathcal{Q} _{\times}^{T} \cap \mathcal{Q} _{\checkmark}^{B}|/|\mathcal{Q} _{\checkmark}^{B}|$, where $\mathcal{Q} _{\checkmark}^{B}$ is the set of Target Domain questions answered correctly at baseline, and $\mathcal{Q} _{\times}^{T}$ is the set of Target Domain questions answered incorrectly in the transfer experiment.
>
> Our results show **no** significant evidence of cross-domain contamination. As shown in the following table, the transfer reality shift rate ($\sigma^T$) was exceptionally low across all pairings, consistently at or below 1%. This demonstrates that the induced Mandela effect is highly domain-related. We thank the reviewer for this insightful question and have added this experiment and analysis to the revised version (**Section 4.3.4** and **Appendix D**).
>
> |$\sigma^{T}$ (%)|Target: History, Time, & Events|Target: Misconceptions & Social Cognition|Target: General Knowledge|Target: Domain-Specific Knowledge|
> |---|---|---|---|---|
> |Source: History, Time, & Events|—|0.8|0|0.1|
> |Source: Misconceptions & Social Cognition|0.9|—|0.2|0.6|
> |Source: General Knowledge|0.4|0.7|—|0.8|
> |Source: Domain-Specific Knowledge|1|0.9|0.6|—|
>
> Again, we thank the reviewer for the valuable feedback. Please let us know if there are any other questions or suggestions.

---

### Author Response · Authors · 2025-12-01
**General Response**

We sincerely thank the Area Chairs and all reviewers for their dedicated coordination and insightful feedback. We have carefully addressed each point and revised the paper accordingly.

We are delighted that the reviewers acknowledged the novelty and significance of our work through the following key contributions:

- **Novelty and Significance (Reviewers BRew, ABwn, LqGM, H3cb):** We present the first systematic study to identify and quantify the Mandela effect in LLM-based multi-agent systems, filling the gap between individual hallucination and group cognitive bias. We reveal that social dynamics amplify false memory formation.
- **Comprehensive Evaluation Benchmark (Reviewers BRew, ABwn, LqGM, H3cb):** We propose MANBENCH to measure the Mandela effect (collective false memory). Reviewers commended our evaluation across diverse models, protocols, and quantitative metrics, confirming that this is a universal vulnerability.
- **Effective Mitigation Strategies (Reviewers BRew, H3cb):** Reviewers recognized our prompt-level and model-level defenses as effective measures to mitigate the Mandela effect. We demonstrate that balanced training enables models to effectively resist misinformation while accepting valid guidance.

In response to the reviewers' concerns, we have provided detailed explanations in the rebuttal and revised the paper. The key updates are outlined below:

- **Generalization in Sensitive Domains (Reviewers ABwn, LqGM, H3cb):** We extended our evaluation to MedMCQA (**Appendix E**), confirming that the Mandela Effect persists in high-stakes domains such as medical diagnosis. Crucially, our defenses remain effective on these unseen tasks, validating their generalization and real-world implications.
- **Theoretical Distinctions (Reviewer LqGM):** We have clarified the scope and theoretical basis of our work.
  - **Differentiation from Related Work:** We updated **Section 2** to explicitly distinguish our focus on collective cognitive bias from traditional faulty agent robustness.
  - **Applicability of Psychological Framework:** We clarified that LLM traits (e.g., sycophancy and memory fragility) do not prevent but rather facilitate the Mandela Effect.
  - **Collaborative Utility:** We clarified in **Section 5.2** why simplistic strategies like "ignoring others" fail, emphasizing the necessity of a cooperative set for valid collaboration.
- **Impact of Agent Heterogeneity (Reviewers LqGM):** We conducted new experiments with heterogeneous groups in **Appendix B**, revealing that heterogeneity amplifies the Mandela effect.
- **Validity of Controlled Interaction Protocols (Reviewers BRew, ABwn):** **Section 6** further clarified that our controlled, multiple-choice format and synthetic protocols are essential for quantifying reality shifts because they provide verifiable ground truth for controlled evaluation.
- **Ratio of Resilience Set and Cooperative Set in Model-level Defense (Reviewer ABwn):** **Appendix C** details the impact of different SFT data ratios, showing that our setting offers a balance between resilience (error resistance) and cooperation (knowledge absorption).
- **Cross-Task Contamination (Reviewer BRew):** **Appendix D** presents the cross-task contamination analysis, confirming that false memories are highly content-dependent and show negligible transferability across different domains.

We appreciate the time and effort Area Chairs and all reviewers have invested in reviewing our work and providing valuable suggestions. We hope that the revised version addresses the concerns raised and contributes to the development of more resilient and reliable LLM-based multi-agent systems.

---

### Meta-Review · Area_Chair_CWSp · 2025-12-05

**Summary:**

This paper examines the Mandela effect, a phenomenon where collective cognitive biases lead multi-agent LLM systems to form shared false memories. It introduces MANBENCH, a benchmark consisting of 4,838 questions across various task domains and interaction protocols to evaluate the prevalence of this effect among 13 LLMs. The findings reveal significant susceptibility to the Mandela effect across all models tested, and the authors propose both prompt-level and model-level mitigation strategies that show up to a 74.4% reduction in its impact, providing insights for enhancing resilience in collaborative AI systems.

**Reviewer Concerns:**

In my opinion, authors adequately addressed nearly all concerns.

Fully addressed:


- *Concerns About Synthetic Dialogues*: Reviewers wished authors had clarified how the synthetic "specious evidence" dialogues map onto real-world scenarios, as the synthetic nature of these narratives might limit their applicability to actual multi-agent deployments. **Authors adequately addressed this by pointing out that this allows them to systematically control the dialogues, and remain psychologically grounded.**


- *Ecological Validity Issues*: A reviewer highlighted that while the MANBENCH tasks adapted from BIG-Bench Hard were valuable, they may not fully capture the complexities of real-world interactions in dynamic, unstructured settings. **Authors addressed this concern, asserting that while ecological validity may be an issue, benchmarks can still be useful.**

- *Generalizability of Defense Mechanisms*: Reviewers felt that the paper lacked evidence of the proposed defense methods’ effectiveness across unseen tasks or specialized domains, leading to uncertainty regarding their broad applicability. **Authors addressed this by conducting additional experiments in medical QA, demonstrating that their defense method generalizes.**

- *Predefined Agent Roles*: Reviewers expressed concern that predetermined roles and consensus direction in the interaction protocols may not accurately represent the nature of real multi-agent systems. **Authors included uncoordinated setups in their work and provided reasoning to address these concerns.**

- *Fundamental Differences Between LLMs and Humans*: Reviewers pointed out that LLMs exhibit fundamentally different characteristics compared to humans, which may affect the study's applicability. **Authors adequately addressed this in their rebuttal, demonstrating an understanding of the distinctions.**

- *Differentiation from Previous Work*: Reviewers questioned the novelty of the evaluation framework compared to existing studies and the potential repurposing of datasets rather than developing truly new challenges. **Authors addressed this by clarifying conceptual and empirical differences from cited works and reinforcing the novelty of their benchmark beyond mere repurposing.**

- *Single LLM Experiments*: Reviewers expressed concerns that conducting experiments with only one LLM per simulation may not align well with the real-world complexities of multi-agent interactions, suggesting the need for heterogeneous LLM setups. **Authors addressed this concern by conducting extra experiments with combinations of different LLMs, amplifying the Mandela vulnerability, thus supporting their claims.**

- *Preliminary Mitigation Strategies*: Reviewers noted that the proposed mitigation strategies rely heavily on predefined rules, which may not generalize well. **Authors acknowledged this and argued that their main contribution lies in the creation of the benchmark and analyses, and that the mitigations are preliminary by design, which is not a real concern.**

- *Discussion of Real-World Risks*: Reviewers suggested that the paper would benefit from a more detailed discussion of practical risks associated with the Mandela Effect in sensitive decision-making contexts. **Authors responded by exploring a medical QA task, which adds a slightly more real-world context to their findings and shows similar effects.**

Possibly not fully addressed:
- *Limited Evaluation Benchmarking*: Two reviewers noted that the benchmark is restricted to multiple-choice questions, which may not adequately reflect the complexity of real-world tasks involving open-ended dialogues or debates. **Authors somewhat addressed this, by arguing that MC benchmarks are still valuable, and pointing to generative evals for future work. I mostly agree with this, though I do believe the MC benchmark can quite easily be measured in a generative way (just don't provide the choices and ask for a free-text response and use LLM-as-judge to determine closest answer match).**

**Reviewer Scores:**

I think most reviewers would have raised their scores, except the one that gave a 2 which seemed pretty negative. However I don't fully agree with this reviewer's concerns.

---

### Decision · Program_Chairs · 2026-01-26

Accept (Poster)